# Deep learning predicts short non-coding RNA functions from only raw sequence data

**Teresa Maria Rosaria Noviello**[1,2], **Francesco Ceccarelli**[3,5], **Michele Ceccarelli**[1,3], **Luigi Cerulo**[2,4] *

**1** Department of Electrical Engineering and Information Technology, University of Naples "Federico II", Napoli, Italy, **2** Biogem Scarl, Istituto di Ricerche Genetiche "Gaetano Salvatore", Ariano Irpino, Italy, **3** CaReBios srl, Ariano Irpino, Italy, **4** Department of Science and Technology, University of Sannio, Benevento, Italy, **5** Computer Laboratory, University of Cambridge, Cambridge, UK

* lcerulo@unisannio.it

**Data Availability Statement:** The data underlying the results presented in the study are available from https://github.com/bioinformatics-sannio/ncrna-deep.

## Abstract

Small non-coding RNAs (ncRNAs) are short non-coding sequences involved in gene regulation in many biological processes and diseases. The lack of a complete comprehension of their biological functionality, especially in a genome-wide scenario, has demanded new computational approaches to annotate their roles. It is widely known that secondary structure is determinant to know RNA function and machine learning based approaches have been successfully proven to predict RNA function from secondary structure information. Here we show that RNA function can be predicted with good accuracy from a lightweight representation of sequence information without the necessity of computing secondary structure features which is computationally expensive. This finding appears to go against the dogma of secondary structure being a key determinant of function in RNA. Compared to recent secondary structure based methods, the proposed solution is more robust to sequence boundary noise and reduces drastically the computational cost allowing for large data volume annotations. Scripts and datasets to reproduce the results of experiments proposed in this study are available at: https://github.com/bioinformatics-sannio/ncrna-deep.

## Author summary

Small non-coding RNAs (ncRNAs) are short non-coding sequences involved in gene regulation in many biological processes and diseases. The lack of a complete comprehension of their biological functionality, especially in a genome-wide scenario, has demanded new computational approaches to annotate their roles. We show that RNA function can be predicted with good accuracy from a lightweight representation of sequence information without the necessity of computing secondary structure features which is computationally expensive. This finding appears to go against the dogma of secondary structure being a key determinant of function in RNA.

This is a *PLOS Computational Biology* Methods paper.

**Funding:** The research leading to these results has received funding from: Associazione Italiana per la Ricerca sul Cancro (AIRC) under the grant number 21846 (grant recipient MC); Ministero dell'Istruzione, dell'Università e della Ricerca PRIN under the grant number 2017XJ38A4-004 (grant recipient MC); and Regione Campania Progetto GENOMAeSALUTE (grant recipient MC). The funders had no role in study design, data collection and analysis, decision to publish, or preparation of the manuscript.

**Competing interests:** The authors have declared that no competing interests exist.

## Introduction

Recent advances in whole transcriptome sequencing have led to the discovery of novel transcribed elements with no apparent functional or protein-coding potential. In the past considered as "dark matter", they are recognized nowadays to play key roles in gene expression regulation in many biological processes and diseases [1]. Several classes of non-coding RNAs (ncRNAs) have been discovered in the last years, stressing on their importance as regulators of cellular development and differentiation. Conventionally, ncRNAs are classified into two major classes according to their length, short (<200 nucleotides) and long (>200 nucleotides) ncRNAs. It is common knowledge that ncRNAs regulate gene expression both on post-transcriptional and transcriptional levels, affect the organization, and modification of chromatin, or have catalytic functions [2]. In particular, short ncRNAs include ribosomal RNAs (rRNAs) and transfer RNAs (tRNAs) involved in mRNA translation, small nuclear RNAs (snRNAs) involved in splicing, small nucleolar RNAs (snoRNAs) involved in the modification of rRNAs, and microRNAs (miRNAs) involved in targeted translational repression and gene silencing. The functional characterization of ncRNAs on a wide scale is currently one of the main challenges of modern genome biology as, compared to protein coding RNAs, they are usually less conserved and expressed.

One of the main efforts in the systematically and automatically classification of ncRNAs have been provided by Rfam database. Rfam [3] is a database that collects ncRNA sequences into families given that, as for protein coding genes, they have evolved from a common ancestor. At the base of each family construction, Rfam starts with at least one experimentally validated example from the published literature with known functional classification. Then, each family is described from a multiple sequence alignment, called seed alignment, that will be used to build a covariance model in order to search for other possible homologous sequences and then expand the family. In this way, Rfam contributes to systematically annotate and analyse ncRNA sequences into families with known function, common ancestor, and, when available, a secondary structure that could provide indications of consensus biological role of that family.

The consolidated evidence that the function of protein coding sequences is strongly associated with the folded secondary and tertiary molecular structure leads to suppose that the secondary structure is a key factor to determine the function of non-coding RNA sequences [4]. Recently, several machine learning approaches have been successfully proven to predict RNA function (according to Rfam classification in families) from secondary structure information. Comparative sequence-based approaches, such as BLAST, are computationally very efficient but exhibit high false negative rates, as they are not able to detect conserved secondary structures. Folding approaches, such as GraPPLE [5], ignore nucleotide composition, are computationally expensive, and incur in high false positive rates, as sequence information is not taken into account. Approaches that combine both structural and sequential information are preferable for a better trade-off between false positives and false negatives. To this aim, INFERNAL adopts a stochastic context-free grammar to capture position-specific conservation and incorporates the RNA secondary structure information directly into the model [6]. A significant improvement with respect to INFERNAL has been obtained with EDeN, a machine learning method that adopts a graph kernel to model the RNA secondary structure input representation [7]. Comparable results have been obtained with nRC, a deep learning approach based on features extracted from secondary structure [8], and RNAGCN, based on a graph convolutional network built on RNA folding data [9].

These methods are all based on the use of structural features to predict ncRNA functions according to their Rfam class. However, the inference of a real secondary structure is still very

challenging and requires high computational cost. Managing this task with well consolidated methods, such as ViennaRNA [10] and iPknot [11], does not prevent the presence of multi step error superimposition, leading to a low prediction accuracy for these methods. Recently, deep learning has emerged as one of the best machine learning approaches for prediction and classification problems in a variety of contexts, such as image or speech recognition, computer design and vision, bioinformatics, and medical image analysis [12, 13]. The greatest strength of using a deep learning approach is that discriminative features, also at high levels of abstraction, can be automatically learned from the input data independently from their nature.

In this paper, we show that small ncRNA function can be predicted with good accuracy just from raw sequence information without the necessity of computing secondary structure features which are known to be computationally demanding. Besides the advantage in terms of computational time, this finding poses a question against the dogma of secondary structure being a key determinant of function in RNA. Evidence shows that with a 3 layer Convolutional Neural Network (CNN) the sequence alone is enough to predict the function of an RNA. Moreover, compared to recent secondary structure based methods, the proposed solution is more robust to sequence boundary noise and is able to reject effectively non-functional sequences. The last two advantages together with fast classification speed are essential for large genome annotation.

CNN has emerged as an approach to extract local feature patterns of high-level abstraction from different and sparsely preprocessed data [14, 15]. Then it is likely that high level functional RNA features are directly learned from sequences by a CNN architecture. How such features are related, if they are, to secondary structure features remains an open question.

## Materials and methods

### Datasets

We compare our deep learning architectures against EDeN, nRC, and RNAGCN, the current state-of-the-art. We do not include INFERNAL as its computational cost is prohibitively expensive and, in literature, it has been shown outperformed by EDeN [7]. We design the evaluation experiments considering two datasets: i) a novel dataset composed of sequences extracted from the Rfam database [3]; and ii) a publicly available dataset of ncRNA sequences distributed among 13 functional macro-classes adopted to evaluate RNAGCN and nRC [9], as the authors of RNAGCN do not provide a publicly available tool.

To build the novel dataset, we started with a set of 650790 sequences distributed among 2570 classes. Sequences encoded with letters different from canonical A, T, C, or G were excluded to simplify computation. This is not a limitation as they constitute a very small subset from the total ($\sim$ 9 out of 1000). To obtain a dataset of only small ncRNA sequences, we excluded classes annotated as long non-coding RNAs and with an average sequence length greater than 200 bases obtaining a dataset with 371619 sequences among 177 Rfam classes. To avoid sequence length dependence we removed classes that can be strongly predicted only by sequence length. To detect such classes, we performed a 10-fold cross validation of a C5.0 decision tree algorithm trained only with sequence lengths. The algorithm performed overall with an average accuracy of 0.46 ($\pm$0.0004) and Kappa statistic of 0.44 ($\pm$0.0004), while per class performance was strongly variable (average F1 measure ranging between 0.07 and 0.95). Four classes, RF00032, RF00436, RF00951, and RF01990, show a strong sequence length dependence (average F1 measure greater than 0.80) and has been removed. This reduced the number of classes to 173 and the total number of sequences by 10% to 333280. Moreover, to make each Rfam class sufficiently representative, we excluded classes with less than 400 samples. This conducted to a final set of 306016 sequences distributed among 88 different Rfam classes

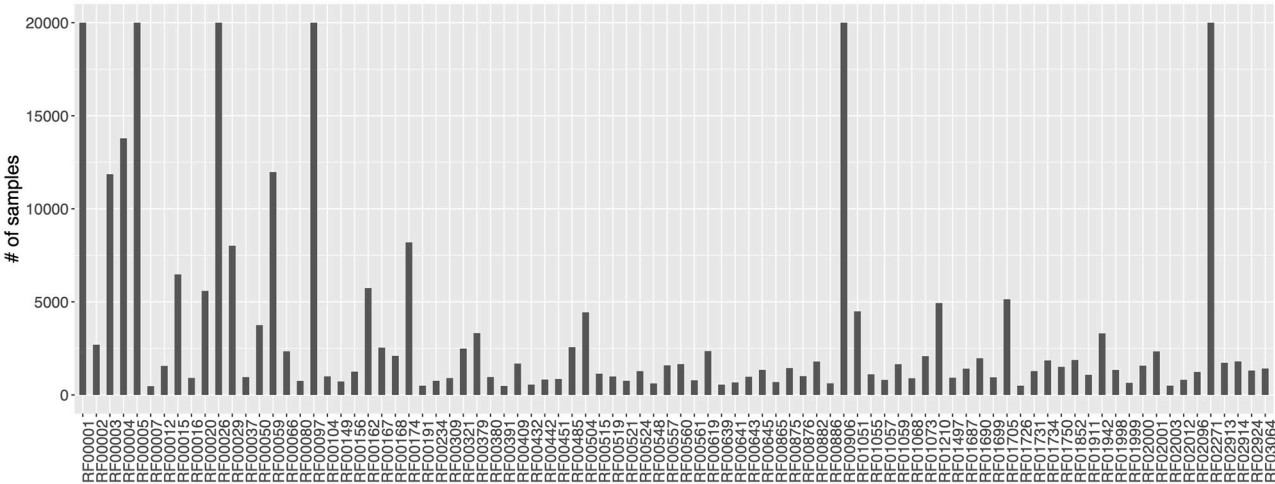

**Fig 1. Distribution of sequences among 88 Rfam classes downloaded from Rfam database.**

(Fig 1). Table 1 shows how Rfam classes are distributed among different non-coding macro classes and Fig 2 shows how sequence lengths are distributed among Rfam classes.

## Input representation of ncRNA sequences

Data representation can strongly affect the performance of machine learning algorithms as they require a good set of hand designed features to work effectively. Instead, the paradigm of deep learning allows in principle to take a simple representation of raw data at the lowest (input) layer that is increasingly transformed into abstract feature representations in subsequent layers. However, as deep learning evolved historically around image analysis, the input of a neural network is typically a matrix which has the intrinsic property to completely preserve pixels locality.

In genomics, as the input is a sequence, typical k-mer representation is able to capture the proximal composition of each nucleotide position. This allows us to learn local patterns of small nucleotide sequence motifs, such as binding sites, but in principle it may not be suited to detect complex spatial patterns of RNA sequences that fold into a 3-dimensional structure,

**Table 1. Distribution of downloaded Rfam classes among non-coding macro classes.**

| non-coding class | Rfam classes |
|---|---|
| snRNA snoRNA | RF00003, RF00004, RF00007, RF00012, RF00015, RF00016, RF00020, RF00026, RF00066, RF00097, RF00149, RF00156, RF00191, RF00309, RF00321, RF00409, RF00432, RF00548, RF00560, RF00561, RF00619, RF01210 |
| Cis-regulatory | RF00037, RF00050, RF00059, RF00080, RF00162, RF00167, RF00168, RF00174, RF00234, RF00379, RF00380, RF00391, RF00442, RF00485, RF00504, RF00515, RF00521, RF00524, RF00557, RF01051, RF01055, RF01057, RF01068, RF01073, RF01497, RF01726, RF01731, RF01734, RF01750, RF02271, RF02913, RF02914 |
| miRNA | RF00104, RF00451, RF00639, RF00641, RF00643, RF00645, RF00865, RF00875, RF00876, RF00882, RF00886, RF00906, RF01059, RF01911, RF01942, RF02000, RF02096 |
| sRNA | RF00519, RF01687, RF01690, RF01699, RF01705, RF02924, RF03064 |
| Intron | RF00029, RF01998, RF01999, RF02001, RF02003, RF02012 |
| rRNA | RF00001, RF00002 |
| tRNA | RF00005, RF01852 |

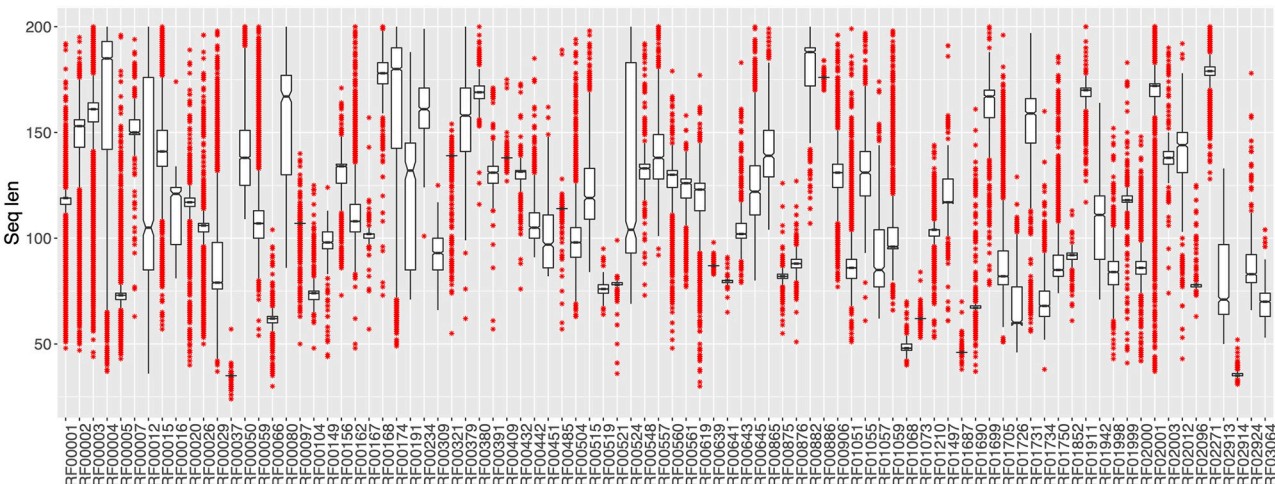

**Fig 2. Distribution of sequence lengths among 88 Rfam classes downloaded from Rfam database.**

where also distant nucleotides could interact. So, it may be necessary to introduce alternative input representations that allow to map linear structures into bi- or three-dimensional structures where such patterns could be detected effectively. Current literature methods basically rely on 3-dimensional secondary structure features predicted using popular RNA folding tools, such as ViennaRNA [10] and iPknot [11]. Although such features have been proven to predict RNA function effectively, they require high computational cost (Table 2). Here, we investigate whether less computationally expensive sequence encodings are sufficient to predict the RNA function. Specifically, we consider $k$-mer and space-filling curves, a lightweight input representation that preserves, almost well, space locality.

$K$-mer encoding is the most common and basic representation of genomic sequence data adopted in deep learning architectures. It consists of associating a binary vector with every consecutive non-overlapping $k$ bases. The vector is all zeros except for the $i$-th entry associated with the unique $k$ word obtained by concatenating $k$ letters from the DNA alphabet (Fig 3). So, for example, a 2-mer encoding of a 100 long sequence produces a sequence of 50 binary vectors of $2^4 = 16$ entries. In our experiments we consider $k$ varying from 1 to 3.

A space-filling curve is a way to traverse a multi-dimensional space of cell elements where every cell is visited exactly once [16]. Thus, a space-filling curve imposes a linear order of points in the multi-dimensional space that can be mapped to a linear sequence of elements. Different space-filling curves have been proposed, each differing in their way of traversing the multi-dimensional space. We consider three types of 2D space-filling curves: Hilbert [17], Morton [18], and Snake (Fig 4). Each cell is then encoded with a four length binary vector of zeros except for the $i$-th entry associated with the unique DNA letter.

**Table 2. Computational cost required to build the input representations of a sequence of length $N$.**

| Input representation | Computational cost | Adopted in |
|---|---|---|
| Hilbert | $O(N\sqrt{N})$ | Noviello *et al.*, 2020 |
| Morton | $O(N\sqrt{N})$ | Noviello *et al.*, 2020 |
| Snake | $O(N)$ | Noviello *et al.*, 2020 |
| $k$–mer | $O(N)$ | Noviello *et al.*, 2020 |
| iPknot | $O(N^5)$ | nRC [8] |
| ViennaRNA | $O(N^7)$ | EDeN [7] and RNAGCN [9] |

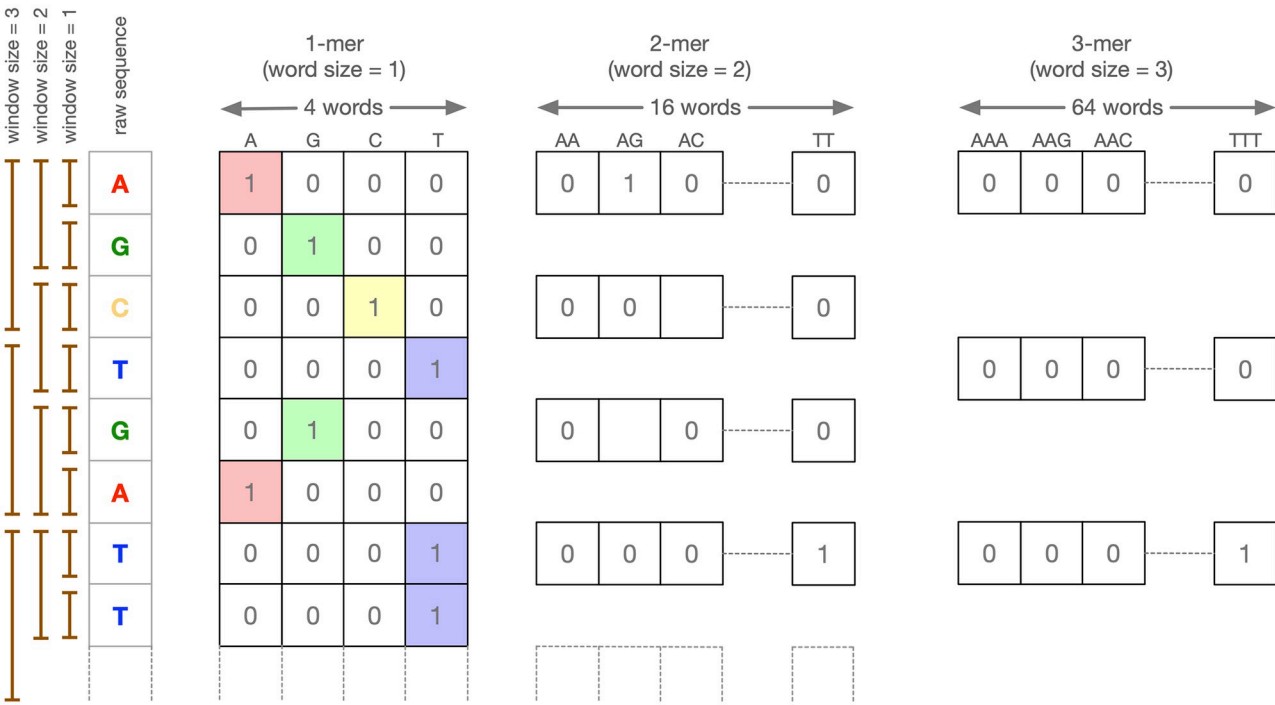

**Fig 3.** *k*-mer representation: Examples of one, two, and tri-mer encodings.

## Deep network architecture

We adopt the standard deep learning CNN architecture depicted in Fig 5. The network is composed of multiple layers of parametrized kernel convolutions, each composed with: a rectified linear unit (ReLU) activation function to reduce the effect of gradient vanishing, a max-

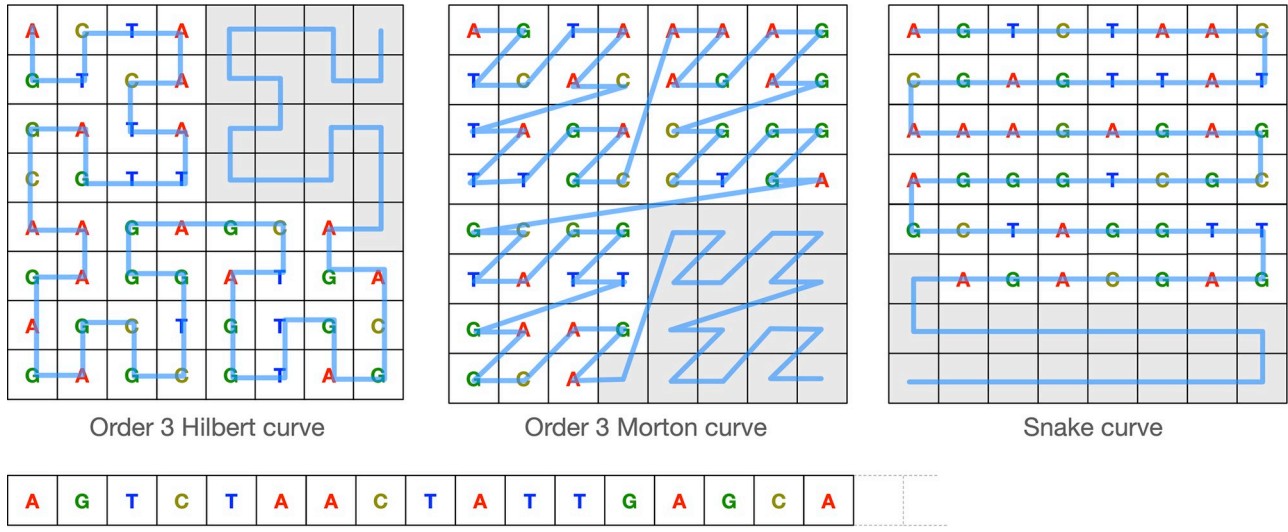

**Fig 4. Examples of bi-dimensional space-filling curves.** The raw linear 47 base long sequence is encoded into the bi-dimensional space-filling curve depicted in blue. The padding necessary to fill the entire space is depicted in grey.

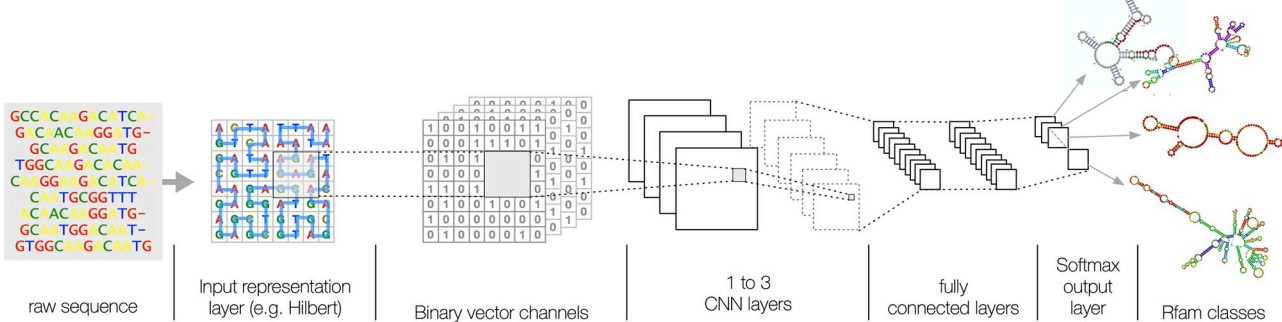

**Fig 5. A graphical representation of the deep learning architecture.** The raw RNA sequence is first encoded into an input layer representation (e.g. Hilbert filling-curve), then up to 3 convolution layers with rectifier activation followed by max-pooling layers perform the learning steps of sub-sequences with functional properties. Finally, two dense layers of rectified linear units are added to reduce data dimension to a softmax multi-class classification output layer.

pooling layer to reduce the size of output, and a 50% drop-out layer to reduce overfitting [19]. We consider an increasing number of CNN layers (ranging from 1 to 3), while the dimension of convolution layers is 1D for k-mer input encodings and 2D for space-filling curve encodings. Input sequence representations are first encoded into binary vectors, where each entry corresponds to a CNN channel, and then padded to the maximum dimension allowed for that representation (Table 3). We consider three padding criteria: i) *random*, where vacant cells are filled with random symbols; ii) *constant*, where vacant cells are filled with a constant symbol drawn from the DNA alphabet; and, iii) *new*, where vacant cells are filled with a new symbol not belonging to the DNA alphabet.

We set empirically the kernel size to 3 and the number of filters at each $i$-th layer to $32 \cdot 2^i$. The architecture is completed with a flatten layer, to turn spatial features into a vector, two dense layers (respectively of 1000 and 500 nodes), and a softmax output to achieve multi-class classification. For the training step, we adopt Adam [20] as optimization algorithm and categorical cross-entropy loss function, suitable for multi-class classification problem [21].

Moreover, in order to have a more comprehensive overview of the most suitable deep learning model for short ncRNA classification, we consider also a Recurrent Neural Network (RNN) architecture in the comparison to the state-of-the-art on the dataset (training and test set) named as "test13" provided by nRC's author [8]. We test three bidirectional Long Short-Term Memory Network (LSTM) RNN architectures with an increasing number of nodes (50,100,150). Since RNNs can process information as sequential data with no predetermined size limit, we apply these architectures on the sequences encoded as $k$-mers with no-padding and not as space-filling curves. Each RNN configuration is composed of a sequence input layer, two bidirectional LSTM layers alternating with two 20% drop-out layers. The architectures are completed with a dense layer and a softmax output for classification.

**Table 3. Maximum dimension allowed for each input representation and sequence of at most 200 nucleotides.**
Dimensions of Hilbert and Morton spaces are the lowest powers of two greater than 200, while the dimension of Snake can be simply obtained consider the ceiling of $\sqrt{200}$.

| Input representation | Maximum dimension allowed |
| --- | --- |
| Hilbert | $16 \times 16$ |
| Morton | $16 \times 16$ |
| Snake | $15 \times 15$ |
| k–mer | 200/k |

## Experiment setup

We considered the ncRNA functional annotation task as a multi-class problem where each class is a collection of functionally related ncRNAs. Accuracy and Kappa statistic are adopted to estimate the overall prediction performance, while per class prediction capability is estimated with weighted F1-measure as more informative in highly unbalanced datasets.

To test for the generalization capacity of the algorithm, we split each Rfam class into three random subsets: train (84%), validation (8%), and test (8%). Validation set was used only to tune the hyper-parameters of the learning algorithm, while test set was used to estimate the predictive performance. To limit the bias due to an over-representation of very similar homologous sequences in random splits, we ensured that for each class all sequences in validation and test sets have a similarity—computed in terms of normalized Hamming distance—less than 0.50 with any other sequence in the training set.

Following the experimental assessment conducted in [7], we assess the prediction performance also under the uncertainty of where ncRNA sequence starts and ends. This could happen, for example, with noise coming from next-generation sequencing. We added to each sequence a varying *boundary noise*, consisting of a random number of nucleotides at the beginning and the end of a sequence preserving the nucleotide and di-nucleotide frequency of the original sequence [22]. We consider the length of the added noise varying among 0%, 25%, 50%, 75%, 100%, 125%, 150%, 175%, and 200% of the original sequence length.

Moreover, we test the rejection capability of the algorithm, *i.e.* the behaviour of the algorithm if presented with non-functional RNA sequences, *i.e* sequences randomly generated by shuffling the initial set and preserving the di-nucleotide composition of each original sequence, or with uncertain sequences. Recently, it has been shown that excluding uncertain samples from test set can drastically improve model performance [23, 24]. To this aim, we adopted Monte Carlo Dropout to estimate the classification uncertainty of a test sample and decide whether to reject or not the sample. We trained the 3 layer CNN architecture with the training set and performed Monte Carlo dropout during test time. Monte Carlo dropout consists to use $N_{mc}$ different dropout versions of the trained model on the same test sample [23]. In each version, $i = 1, \ldots, N_{mc}$, a random set of nodes is deleted allowing to obtain a discrete probability distribution $p_{ik}$ among all class values, $k = 1, \ldots, C$. From such a distribution the uncertainty of classification can be estimated in different ways [23, 24].

In our experiment we adopted $N_{mc} = 50$ and evaluated two uncertainty estimators: *Information Entropy* and *Top Difference*. Information entropy is defined as:

$$ H = -\sum_{k} p_k log_2(p_k + \epsilon) $$

where $p_k = \frac{1}{N_{mc}} \sum_{i} p_{ik}$, is the mean over all predicted probabilities for a class $k$, and $\epsilon$ is added for numerical stability. The *Top Difference* is defined as the difference between the two top, in average most probable, predicted classes $k_1$ and $k_2$, calculated as:

$$ D = p_{k_1} - c\sigma_{k_1} - (p_{k_2} + c\sigma_{k_2}) $$

where $\sigma_k$ is the standard deviation of the discrete probability distribution $p_{ik}$ among $i$, and $c$ a constant we set to $c = 0.6$.

We evaluated the capability to predict functional vs. non-functional RNA sequences plotting the ROC curve of each estimator on a doubled test set obtained by adding to each sequence of the original test set a shuffled version preserving di-nucleotides distribution. As an example, we evaluated the gain in classification performance on the original test set where

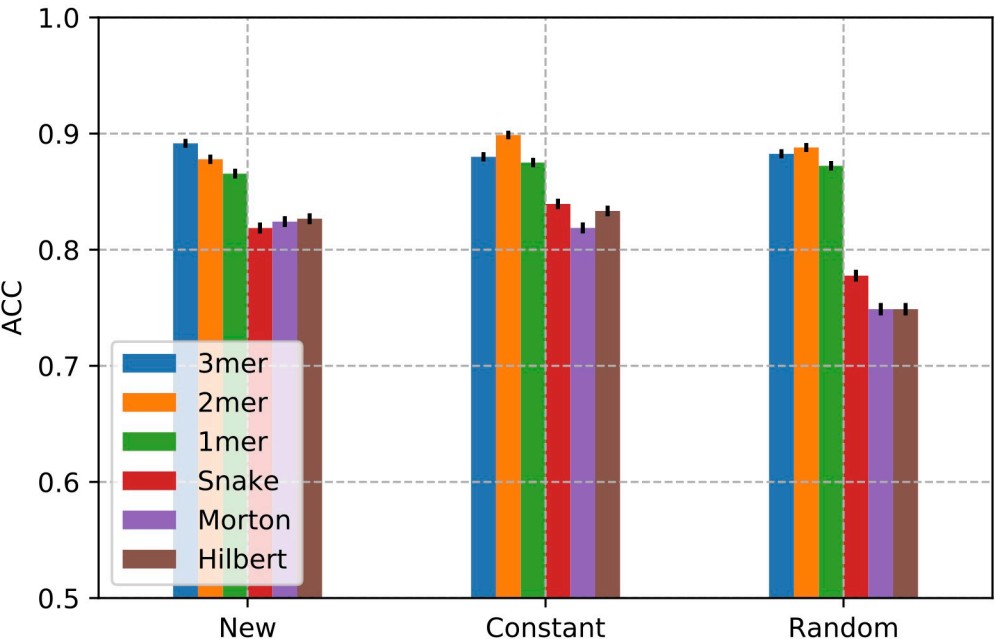

**Fig 6. Classification performance in term of Accuracy obtained in the test set with different padding schemas.**
The deep learning architecture is composed by 3 CNN layers. Confidence intervals are drawn assuming a normal distribution of classification error.

uncertain sequences are filtered out considering a decision threshold estimated empirically. We found the following best thresholds for the estimators: $H > \frac{1}{3}log2\left(\frac{1}{C}\right)$ for the Information Entropy estimator and $D < 0$ for Top Difference estimator.

## Results and discussions

### Padding with random symbols affects space-filling curve performance

At first, we evaluated the impact on classification performance of different input sequence representations and padding criteria. To this aim, we adopted a 3 CNN layer architecture and evaluated the prediction performance against the novel Rfam dataset. Fig 6 shows the obtained results, in terms of Accuracy (ACC). *K*-mer encodings are not sensitive to padding criteria, while space-filling curve encodings exhibit a significant accuracy drop ($\sim$ 10-15%) with random padding.

Having in proximity both distal and close elements of a sequence constitutes a disadvantage when inputs are filled with random padding, while constant and new symbol padding is less prone to affect overall prediction performance.

### CNN number of layers contributes to performance improvement

Fig 7 shows the impact of neural network depth, codified with the number of CNN layers, on the classification performances. A number of CNN layers equal to zero corresponds to a dense network. According to the above results, a new symbol has been used as padding criteria and performances were evaluated against the novel Rfam dataset.

As expected, the absence of CNN layers strongly affects the learning step resulting in a low rate of accuracy for all the tested input representations. In a dense network, fully connected layers see the data as 1D vectors so it is likely that high level (spatial) relationships and local

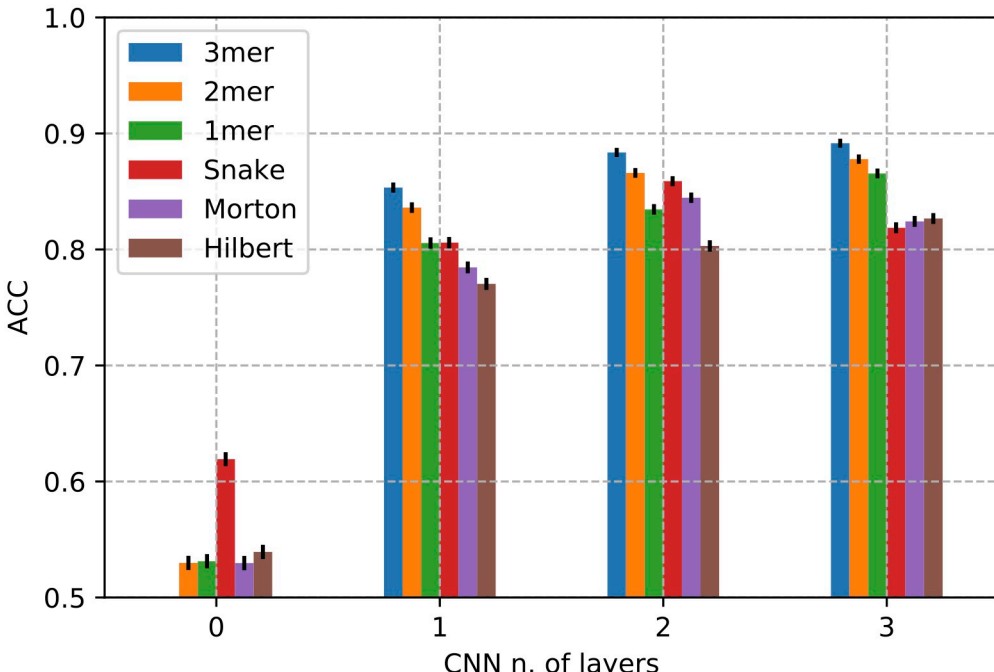

**Fig 7. Classification performance in term of Accuracy obtained in the test set with different number of layers using CNNs where inputs are padded with a new symbol.** Zero indicates a dense network. Confidence intervals are drawn assuming a normal distribution of classification error.

patterns are not captured. Conversely, increasing architecture's depth enhances, almost linearly, the learning process of high-level abstract and spatially localized features supposedly connected to RNA function. Adding just one CNN layer increases the prediction accuracy by two-fold, advancing it to the range 0.80–0.90 for all input representations. A significant increment is registered for k-mer while adding more layers to space-filling curve representations does not significantly affect the prediction performance. The use of space-filling curves as a proxy for modelling long-range interactions between nucleotides show the worst performance. This does not exactly dismiss the importance of structural effects but poses a question on the necessity to go through the RNA structure to learn RNA functions.

## K-mer encodings are more robust to boundary noise

Fig 8 shows the impact of boundary noise on classification performances against the novel Rfam dataset for each considered input sequence representations. The comparison, in terms of accuracy, with state-of-the-art methods, EDeN and nRC, is also shown. According to the above results, a new symbol has been used as padding criteria and three CNN layers as the depth of the architecture. At 0% of boundary noise, i.e. original sequences without noise addition, all considered input data representations reach the highest levels of accuracy. EDeN and nRC show performances similar to k-mer representations (0.87–0.90), while spatial curve representations exhibit an accuracy ranging between 0.82 and 0.83. Increasing the percentages of boundary noise a decrement of performance is registered for all methods. The decrement is more slightly for $k$-mer representations and more prominent for spatial-curves and the state-of-art methods, EDeN and nRC. At 200% of boundary noise the performance, in terms of accuracy, of $k$-mer representations are in the range 0.81–0.84, while for all others the performance drops in the range 0.64–0.70.

 

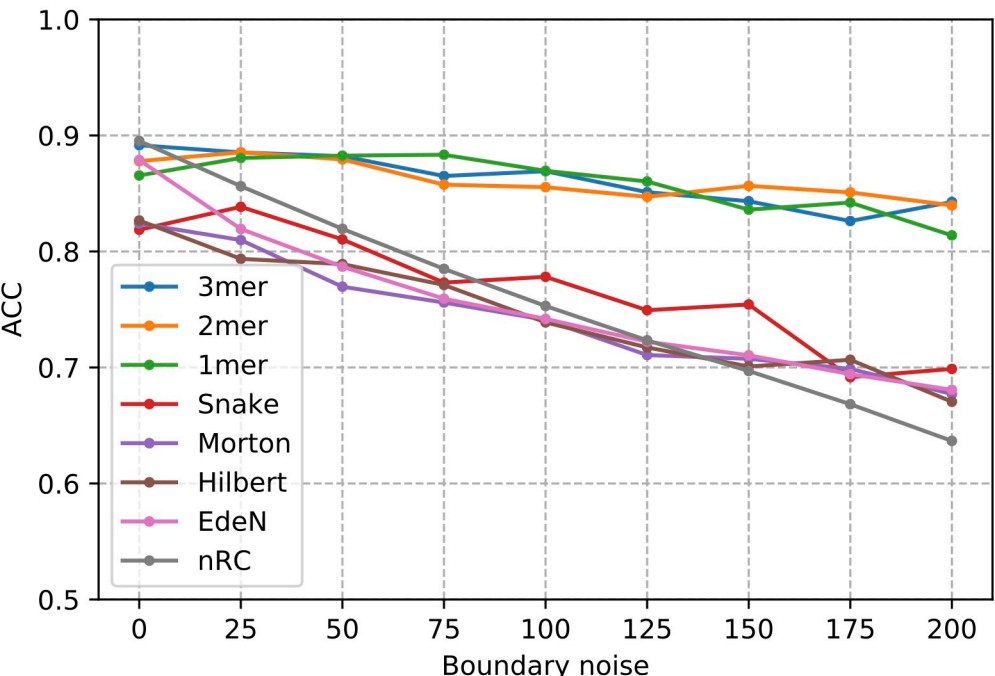

**Fig 8. Classification performance in term of Accuracy obtained in the test set at different boundary noise levels.** The deep learning architecture is composed by 3 CNN layers and inputs are padded with a new symbol.

S1 and S2 Tables show the breakdown of the classification performances of a 3 CNN layers architecture at class level in terms of F1-measure (F1), and their macro and weighted averages, respectively with the minimum and the maximum noise level (0% and 200%). At 0% noise level all methods show almost similar performances, in terms of both weighted and macro F1 averages. Instead, at 200% noise level, $k$-mer representations show better performances, exhibiting a decrement around 7% for both weighted and macro F1 averages. The state of the art, EDeN and nRC, are affected by a drop ranging between 25% and 50%, while the performance reduction of spatial curves is attested between 12% and 24%.

At 200% noise level a high concordance of per class performance can be observed within the $k$-mers group, between EDeN and nRC, and within spatial-filled curves. There are 24 classes where all methods are wrong (*i.e.* F1 less than 0.60) in a similar way and 40 classes where the F1 measure of $k$-mer representations are in average 50% higher than others in literature methods. The state-of-art outperforms k-mer representations in only 6 classes (average F1-measure 50% higher).

## Monte Carlo Dropout robustly recognizes non-functional RNA sequences and improves prediction performance on non-rejected sequences

Fig 9 shows the performance, estimated in terms of Area under ROC, of rejecting non-functional RNA sequences of two classification uncertain estimators, Information Entropy and Top Distance. Both estimators exhibit similar performance, 0.92 for Information Entropy and 0.90 for Top Distance.

Table 4 and Fig 10 show, respectively, the overall and per class performance after Monte Carlo Dropout of uncertain samples encoded with no boundary noise. Overall performance is calculated in terms of Accuracy, Kappa statistic, and Matthew Correlation Coefficient (MCC),

 

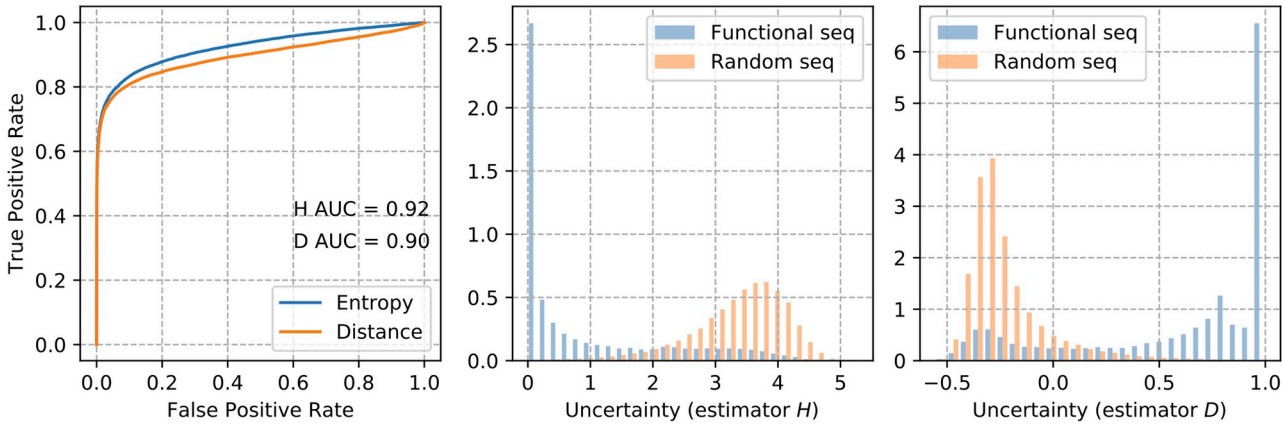

**Fig 9. Recognizing non-functional RNA with Monte Carlo Dropout.** Sequences are encoded with 1-mer and performance is estimated in terms of Area under ROC (on the left). Figures on the right shows the distributions of functional and non-functional RNA sequences among Information Entropy (H) and Top Distance (D).

while per class performance is calculated in terms of F1-measure. The percentage of dropped samples for each class and the overall percentage of dropped samples are also shown.

For all input representations, an overall increment of accuracy can be registered. Comparing results reported in Table 4 with S1 Table the following increments can be observed for Information Entropy, 3-mer 11.23%, 2-mer 12.50%, 1-mer 13.79%, Snake and Morton 19.51%, Hilbert 18.07%; and the following for Top Distance, 3-mer 10.11%, 2-mer 11.36%, 1-mer 13.79%, Snake 18.29%, Morton 17.07%, Hilbert 16.86%. The highest percentage of dropout samples is registered for Sanke with Information Entropy (41.04%), while the lowest is registered for 1-mer with Top Distance (18.61%)

For Information Entropy, the worst per class performance (i.e. F1 less than 0.60) is registered for a number of classes ranging between 8 and 13; instead, for Top Distance the number of worst performing classes ranges in 8–11 for k-mer and 11–18 for spatial curves. For Information Entropy a strong improvement of per class performance (i.e. F1 50% higher) is registered for a number of classes ranging between 23–26 for k-mer and 28–34 for spatial curves, while for Top Distance the number of classes ranges in 24–26 for k-mer and 26–30 for spatial curves.

**Table 4. Overall performance improvement, in terms of Accuracy, Kappa, and MCC, after Monte Carlo Dropout of uncertain samples encoded with 1-mer.**

| Estimator | Approach | Accuracy | Kappa | MCC | % of rejected samples |
|---|---|---|---|---|---|
| Entropy | 3mer | 0.99 | 0.99 | 0.99 | 24.28 |
| | 2mer | 0.99 | 0.99 | 0.99 | 24.32 |
| | 1mer | 0.99 | 0.99 | 0.99 | 18.79 |
| | Snake | 0.98 | 0.98 | 0.98 | 41.04 |
| | Morton | 0.98 | 0.97 | 0.97 | 38.50 |
| | Hilbert | 0.98 | 0.98 | 0.98 | 37.76 |
| Top | 3mer | 0.98 | 0.98 | 0.98 | 20.07 |
| | 2mer | 0.98 | 0.98 | 0.98 | 21.55 |
| | 1mer | 0.99 | 0.99 | 0.99 | 18.61 |
| | Snake | 0.97 | 0.97 | 0.97 | 31.17 |
| | Morton | 0.96 | 0.96 | 0.96 | 31.69 |
| | Hilbert | 0.97 | 0.97 | 0.97 | 31.09 |

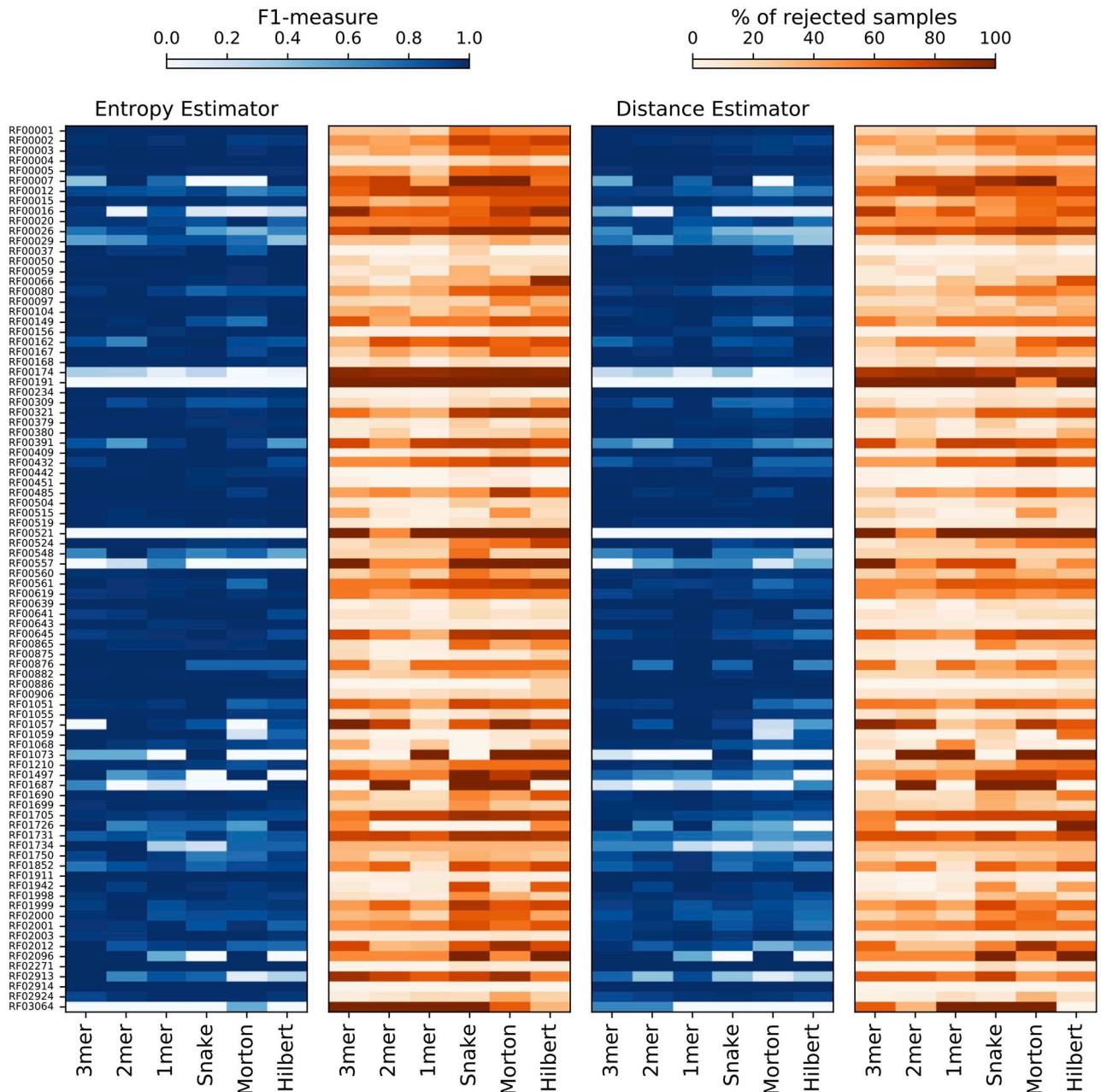

**Fig 10. The effect on per class prediction performance of rejecting uncertain samples (1-mer encoded) with Monte Carlo Dropout.**

## Comparison with RNAGCN

The recent proposed RNAGCN method, based on a graph convolutional network, is evaluated against a dataset where ncRNA sequences are classified over 13 functional macro-classes [9]. As the authors of this method do not provide an executable tool, we were not able to evaluate the proposed method against our novel Rfam dataset. So, we evaluated our approach against the publicly available datasets which were originally adopted by the authors of nRC [8]. In particular, we choose the dataset called *test13* which is the one that shows the best RNAGCN results.

**Table 5. Summary of results on the dataset called *test13* containing 13 non-coding classes.** Results for nRC and RNAGCN are taken from [9].

| Architecture | Approach | Accuracy | Recall | Precision | F1-score | MCC |
|---|---|---|---|---|---|---|
| | EDeN | 0.67 | 0.60 | 0.75 | 0.65 | 0.61 |
| | nRC | 0.82 | 0.82 | 0.81 | 0.82 | 0.80 |
| | RNAGCN | 0.86 | 0.86 | 0.86 | 0.86 | 0.85 |
| RNN 50 nodes | 1mer | 0.86 | 0.86 | 0.86 | 0.86 | 0.85 |
| | 2mer | 0.77 | 0.77 | 0.77 | 0.77 | 0.75 |
| | 3mer | 0.77 | 0.77 | 0.77 | 0.77 | 0.75 |
| RNN 100 nodes | 1mer | 0.88 | 0.88 | 0.88 | 0.87 | 0.87 |
| | 2mer | 0.79 | 0.79 | 0.79 | 0.79 | 0.77 |
| | 3mer | 0.78 | 0.78 | 0.79 | 0.78 | 0.75 |
| RNN 150 nodes | 1mer | 0.89 | 0.90 | 0.90 | 0.90 | 0.89 |
| | 2mer | 0.80 | 0.80 | 0.80 | 0.79 | 0.78 |
| | 3mer | 0.79 | 0.79 | 0.79 | 0.79 | 0.77 |
| CNN standard | 1mer | 0.88 | 0.88 | 0.89 | 0.88 | 0.87 |
| | 2mer | 0.83 | 0.83 | 0.84 | 0.83 | 0.82 |
| | 3mer | 0.81 | 0.81 | 0.82 | 0.81 | 0.79 |
| | Morton | 0.78 | 0.78 | 0.79 | 0.78 | 0.77 |
| | Snake | 0.82 | 0.82 | 0.83 | 0.81 | 0.80 |
| | Hilbert | 0.81 | 0.81 | 0.84 | 0.82 | 0.80 |
| CNN improved | 1mer | **0.96** | **0.96** | **0.96** | **0.96** | **0.96** |
| | 2mer | 0.92 | 0.92 | 0.92 | 0.92 | 0.91 |
| | 3mer | 0.88 | 0.88 | 0.88 | 0.88 | 0.86 |
| | Morton | 0.86 | 0.86 | 0.88 | 0.86 | 0.85 |
| | Snake | 0.86 | 0.86 | 0.88 | 0.86 | 0.85 |
| | Hilbert | 0.86 | 0.87 | 0.89 | 0.87 | 0.86 |

Table 5 reports the results obtained. Surprisingly EDeN exhibits a performance that is significantly lower, in terms of Accuracy, than those obtained against the novel Rfam dataset. Instead with both a standard CNN architecture (Fig 5) and different configuration of Bidirectional LSTM RNNs we obtained performances almost similar to RNAGCN and nRC and almost consistent with those obtained in previous experiments. To test if there is room of improvement, we explored alternative CNN architectures. After a thorough empirical evaluation of different architectures, we obtained an improved configuration that shows an increment between 5% and 10%, in terms of Accuracy, with respect to the standard architecture adopted in previous experiments (Table 5). The improved architecture is composed of 5 CNN layers interleaved with batch normalization, Leaky ReLU activation, and max-pooling. GaussianNoise to reduce overfitting is added every 2 CNN layers and a dropout rate at 20% is added after the last CNN layer. The network is completed with two dense layers, respectively of 128 and 64 units, to reduce input dimensions, and a final softmax layer for the output class. AMSGrad optimization [25], with a learning rate at 0.0005, has been adopted in the learning step.

## Conclusion

Recent advances in high throughput technologies have allowed the discovery of a large number of novel transcript elements, called ncRNAs, and previously considered to lack functional potential. ncRNAs represent a very heterogeneous group of RNA in terms of their length, biogenesis, and functions which can be divided into long non-coding RNAs and short non-

coding RNAs. Due to their complex nature, great challenges still remain for reaching a full comprehension of ncRNAs, demanding the development of computational approaches able to detect and annotate their biological functions according to family identity.

In this work, we proposed a deep learning approach to classify short ncRNA sequences into Rfam classes. A comparative assessment with the state-of-the-art graph kernel methods shows that the deep learning approach is more robust to boundary noise when k-mer input representations are adopted. CNN number of layers contributes to performance improvement while random padding schema affects negatively space-filling curve performance. The deep learning architecture allows for less computational cost input representations than sequence-structure input representations of graph kernel methods. This allows for the classification of large scale genomic data and poses an interesting question against the dogma of secondary structure being a key determinant of function in RNA.

Moreover, since both standard configurations of CNN and RNN architectures are suitable for inferring the functionality of RNAs from their sequences without structural information, a hybrid approach with both CNN and RNN layer blocks could represent the best choice and improve the classification performance. Ideally, a first convolutional layer block could identify short sequence motifs correlated with the biological role of the short ncRNA family, and then a recurrent layer block could learn long-term relationships between inferred functional motifs.

The empirical evaluation of deep learning models in this study, especially regarding CNNs, let us suppose that abstract features associated with RNA functions are effectively learned from simple input representations (i.e. k-mer) and that any further structural encoding in the input representation, such as those carried by space-filling curves, does not contribute to performance improvement. To what extent such features are related to secondary structure features remains an open question.

## Supporting information

**S1 Table. Per class performance evaluated with a 3 layer CNN network, new padding symbol, and 0% of boundary noise.**
(XLSX)

**S2 Table. Per class performance evaluated with a 3 layer CNN network, new padding symbol, and 200% of boundary noise.**
(XLSX)

## Author Contributions

**Conceptualization:** Teresa Maria Rosaria Noviello, Michele Ceccarelli, Luigi Cerulo.

**Data curation:** Teresa Maria Rosaria Noviello.

**Formal analysis:** Teresa Maria Rosaria Noviello, Francesco Ceccarelli, Luigi Cerulo.

**Funding acquisition:** Michele Ceccarelli.

**Investigation:** Teresa Maria Rosaria Noviello, Francesco Ceccarelli, Luigi Cerulo.

**Methodology:** Teresa Maria Rosaria Noviello, Francesco Ceccarelli, Luigi Cerulo.

**Software:** Teresa Maria Rosaria Noviello, Francesco Ceccarelli, Luigi Cerulo.

**Supervision:** Michele Ceccarelli, Luigi Cerulo.

**Writing – original draft:** Teresa Maria Rosaria Noviello, Francesco Ceccarelli, Michele Ceccarelli, Luigi Cerulo.

**Writing – review & editing:** Teresa Maria Rosaria Noviello, Francesco Ceccarelli, Michele Ceccarelli, Luigi Cerulo.

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
