## [Decision Letter · Decision Letter 0]

12 Jul 2020

Dear Dr. Cerulo,

Thank you very much for submitting your manuscript "Deep learning predicts non-coding RNA functions from only raw sequence data" for consideration at PLOS Computational Biology.

As with all papers reviewed by the journal, your manuscript was reviewed by members of the editorial board and by several independent reviewers. In light of the reviews (below this email), we would like to invite the resubmission of a significantly-revised version that takes into account the reviewers' comments.

We cannot make any decision about publication until we have seen the revised manuscript and your response to the reviewers' comments. Your revised manuscript is also likely to be sent to reviewers for further evaluation.

Sincerely,

Ilya Ioshikhes

Associate Editor

PLOS Computational Biology

Weixiong Zhang

Deputy Editor

PLOS Computational Biology

Reviewer's Responses to Questions

**Comments to the Authors:**

Reviewer #1: 1. This paper proposes a deep learning method to predict the classes of some non-coding RNAs. It is a little misleading to say “Deep learning predicts non-coding RNA functions”. Besides the well-known classes of non-coding RNAs, the proposed method may not be used to predict the functions of most non-coding RNAs such as lncRNAs, circRNAs, etc.

2. It is unclear why the authors “removed sequences greater than 150 bases” as most non-coding RNAs are longer than 200 nucleotides. It is also unclear how sequence redundancy in the dataset was handled.

3. While k-mer compositions are widely used for input encoding of sequence, they do not capture all the sequential information. Strictly speaking, the k-mer features are not “raw sequence data”. These k-mer features may work well for some ncRNA classes, but not for others. Also, since RNA sequence determines structure that often underlies function, it is unclear how “this finding poses a question against the dogma of secondary structure being a key determinant of function in RNA”.

4. The deep learning architecture in this study used convolutional neural network (CNN). I wonder whether some other deep learning techniques, such as recurrent neural network and word embedding, were also tested for this problem.

5. What are “non-functional RNA sequences”? The RNA sequences that do not belong to the considered classes can also be biologically functional.

6. The classifier developed in this study should be compared directly with the previous models, especially the ones using RNA structural features. The results for nRC and RNAGCN in Table 6 were taken from a previous study. It is unclear whether the same datasets and testing strategy were also used in the previous study.

Reviewer #2: Summary

In recent years, there has been research evidence that secondary structure is the key factor to know the function of RNA. Some machine learning based methods have been successfully proved to be able to predict RNA function from secondary structure information. At present, there are more or less deficiencies in the existing methods for predicting RNA function on the market, such as BLAST, which has a high false negative rate, GraPPLE, which has a high false positive rate, and INFERNAL, which has a high computational cost. In this case, the author proposes a method based on the original sequence without calculating the known secondary structure features. The method is more robust to the sequence boundary noise and reduces drastically the computational cost allowing for large data volume annotations. The last two advantages together with fast classification speed are essential for large genome annotation.

Major Comments

In general, the idea of this paper is to find a new way to predict RNA function from the original sequence information instead of the existing methods of predicting RNA function through secondary structure, which is of great significance. However, when using k-mer and space filling curve to represent input, the author can add some improvements to these two existing methods to some extent. Secondly, two uncertainty estimators, information entropy and top difference, were evaluated in the prediction of RNA function. For the two uncertain estimators threshold setting, the author lacks the corresponding information. Finally, in assessing RNA function, the author assumes that any further structural coding in the input representation does not help improve performance, which remains to be debated and requires corresponding arguments to prove.

Minor Comments

Picture layout: The graphs and tables in the paper are far apart from the content of the text that concerns them and it seems very inconvenient.

Supplementary Notes: (13th line from the bottom, page 4) Sentence “In our experiments we consider k varying from 1 to 3” needs to be supplemented to explain why k varies from 1 to 3 and the effect of K on the experiment.

Subjective argument: (1st line from the bottom, page 5) The sentence “We set empirically the kernel size to 3 and the number of filters at each i-th layer to 32 * 2i” is too subjective in a sense and the author should make it clear what experience the size of the kernel and the number of filters at each i-th layer are based on.

Reviewer #3: The article by Noviello et al. is a nice investigation on non-coding RNAs for which functional annotations are beneficial for the

biological community. The authors exploited deep learning methods to tackle the challenge and their results shed new light on the structure-function relationship in this class of biomolecules.

The authors also provided all the scripts and documentation to reproduce their work and compared their work to other state-of-the-art methods.

The work is nicely written and logical to follow, I have only minor comments to be addressed:

1. a general proofreading to get rid of the remaining typos and some grammatical errors, or too wordy sentences

2. I am not an expert on non-coding RNAs and I was wondering in reading about the dataset curation how the 41 classes have been selected and in general to know more about how the classification of non-coding RNA sequences in classes is done

This might be beneficial also for a broad audience as the one of PLOS COMP BIOL.

3. It will be nice if the authors could explain a little bit more the rationale behind the choice of the deep network architecture to this case study instead than other approaches also to benefit a broader audience

4. make the conclusions less technical and more accessible to biologists so that they can really appreciate the value of the work

**Have all data underlying the figures and results presented in the manuscript been provided?**

Reviewer #1: Yes

Reviewer #2: None

Reviewer #3: Yes

PLOS authors have the option to publish the peer review history of their article (what does this mean?). If published, this will include your full peer review and any attached files.

Reviewer #1: No

Reviewer #2: No

Reviewer #3: **Yes: **Elena Papaleo
---

## [Decision Letter · Decision Letter 1]

28 Sep 2020

Dear Dr. Cerulo,

We are pleased to inform you that your manuscript 'Deep learning predicts short non-coding RNA functions from only raw sequence data' has been provisionally accepted for publication in PLOS Computational Biology.

Best regards,

Ilya Ioshikhes

Associate Editor

PLOS Computational Biology

Weixiong Zhang

Deputy Editor

PLOS Computational Biology

Reviewer's Responses to Questions

**Comments to the Authors:**

Reviewer #1: The authors did a good job in addressing my concerns.

Reviewer #2: The paper is very much improved and I have no problem in recommending it for publication.

**Have all data underlying the figures and results presented in the manuscript been provided?**

Reviewer #1: Yes

Reviewer #2: None

PLOS authors have the option to publish the peer review history of their article (what does this mean?). If published, this will include your full peer review and any attached files.

Reviewer #1: No

Reviewer #2: No

---

## [Editor Report · Acceptance letter]

3 Nov 2020

PCOMPBIOL-D-20-00903R1 

Deep learning predicts short non-coding RNA functions from only raw sequence data

Dear Dr Cerulo,

I am pleased to inform you that your manuscript has been formally accepted for publication in PLOS Computational Biology. Your manuscript is now with our production department and you will be notified of the publication date in due course.

With kind regards,

Nicola Davies
